# Righting the Misperceptions of Men Having Sex with Men: A Pre-Requisite for Protecting and Understanding Gender Incongruence in Vietnam

**DOI:** 10.3390/jcm8010105

**Published:** 2019-01-17

**Authors:** Van Anh T. Nguyen, Ngoc Quynh H. Nguyen, Thu Hong Khuat, Phuong Thao T. Nguyen, Thu Trang Do, Xuan Thai Vu, Kien Tran, Manh Tung Ho, Hong Kong T. Nguyen, Thu Trang Vuong, Quan Hoang Vuong

**Affiliations:** 1Institute for Social Development Studies (ISDS), Hanoi 100000, Vietnam; nguyentva@gmail.com (V.A.T.N.); ween.nguyen@gmail.com (N.Q.H.N.); hongisds@gmail.com (T.H.K.); ng.thaonp@gmail.com (P.T.T.N.); ttrang.do@gmail.com (T.T.D.); vuxuanthai160382@gmail.com (X.T.V.); trankien@vnu.edu.vn (K.T.); 2School of Law, Vietnam National University, Hanoi 100000, Vietnam; 3Centre for Interdisciplinary Social Research, Phenikaa University, Yen Nghia, Ha Dong, Hanoi 100803, Vietnam; 4Faculty of Economics and Finance, Phenikaa University, Yen Nghia, Ha Dong, Hanoi 100803, Vietnam; 5Vuong & Associates Co., Hanoi 100000, Vietnam; htn2107@caa.columbia.edu; 6Sciences Po, 75337 Paris, France; thutrang.vuong@sciencespo.fr

**Keywords:** men who have sex with men, MSM, transgenders, LGBTIQ rights, clinical process, Vietnam

## Abstract

Protecting the rights of the lesbians, gays, bisexuals, transgender, intersex, and queers (LGBTIQ) population requires, first and foremost, a proper understanding of their sexual orientation and gender identity. This study highlights a severe misunderstanding and lack of knowledge among health professionals in Vietnam with regard to the men who have sex with men (MSM) and transgenders. This study uses (i) a survey based on the convenience sampling method among 150 health workers that covered 61 questions and (ii) 12 in-depth interviews in two metropolitan centres in Vietnam, Hanoi and Ho Chi Minh city. Three main topics are explored: (i) the general knowledge of healthcare workers about MSM and transgenders; (ii) their knowledge about the sexual reproductive health and human immunodeficiency virus/acquired immunodeficiency syndrome (HIV/AIDS) risks of MSM and transgenders; and (iii) their attitudes and behaviors towards MSM and transgenders. One of the notable findings is how prevalent the misperceptions are across the board, namely, in staff of both sexes, in both cities, at various kinds of medical facilities, at different work positions and educational levels. Half of the respondents consider transgenders to have a curable mental problem while 45% say MSM only have sex with males. Most remarkably, 12.7% state if they have any choice, they want nothing to do with MSM and transgenders. The study finds there is a considerable percentage of health professionals who lack knowledge about the diversity of sexual orientation, gender identity, and health issues related to the sexual minorities and gender non-conforming population. To improve the clinical process for serving these at-risk groups, the study suggests the continual education for the health workers needs to be added to their formal as well as in-job training.

## 1. Introduction

In a landmark study in *The Lancet* in 2016, Reisner et al. [1] noted the substantial gaps in empirical research on the global health burden of transgender people. Upon a review of data from the peer-reviewed scientific literature on the topic in 30 countries, the team confirmed that transgender people are indeed surrounded by adverse health risks across high-, middle-, and low-income settings [1]. To improve scholarly understanding of the topic, this study, through the angle of health professionals, examines the specific lack of understanding that prevent the men who have sex with men (MSM) and transgender individuals in two metropoles of Vietnam from accessing quality healthcare. Given that sexual and gender minority groups, such as MSM and transgender individuals, represent vulnerable populations at increased risk for human immunodeficiency virus (HIV) infection, addressing their needs and concerns is an urgent matter not only for them but also for the sake of the social public health system. The term MSM is used to highlight the distinction between sexual identity issue and sexual behavior issue [2,3], while the term transgender describes people whose assigned sex at birth differs from their current gender identity or expression [1]. A transgender person is either a male-to-female or female-to-male individual who may engage in heterosexual or homosexual sex [4].

Vietnam, a lower middle-income country that in 2017 effectively legalized gender reassignment and reaffirmation in its revised Civil Code [5], has joined the list of some 20 countries to have passed some form of legislation recognizing the rights of transgender people [6]. The move tops a series of progressive steps Vietnam has taken in recent years concerning the sexual and gender non-conforming groups, which include repealing the heteronormative definition of marriage and abolishing a ban on same-sex marriage (without outright accepting it yet) [7]. At the same time, Vietnamese society is described as based on “strict sexual dimorphism” and heteronormativity and thus, remains largely intolerant of a third gender and homosexuality [8]. This means the lesbian, gay, bisexual and transgender (LGBT) individuals still face “widespread abuse and discrimination, particularly in their homes,” as some gay rights activists reported [9,10]. The underlying reason is the incompatibility of homosexuality and the Confucian virtue of filial piety, which expects a man to have offspring and carry on ancestral worship. The deviation of transgenders from traditional rigid gender stereotypes is also another reason for this group to be not accepted. Therefore, to understand the depth of their difficulties, the following sub-sections will review the extant literature on the major barriers facing the MSM and transgender population worldwide and in Vietnam.

### 1.1. Global Health Burdens of Men Who Have Sex with Men (MSM) and Transgenders

Studies throughout the world have confirmed that misperceptions often lie at the heart of the mistreatment of people who identify as transgender or gender non-conforming both in clinical care and other social contexts [11,12,13,14,15,16,17,18]. For instance, although transgender men who have female reproductive organs and genitalia need gynecologic care, a study has shown that most of them are reluctant to visit a gynecologist annually due to gender identity concerns [15]. Similar experiences were reported among the MSM in Malaysia [14], transgender youth in South Africa [12] and Jamaica [16], transgender women in Peru [17] and in sex-segregated jails and prisons in the United States [18], to name a few. The mistreatment manifests in various forms, whether that be outright verbal harassment, discrimination, confidentiality breaches, violence, or simple negligence and insensitive comments. Such behaviors and attitudes not only prevent transgender people from accessing normal medical services but also drive a wedge between them and society. Indeed, when looking at the health outcomes of transgender people, researchers have mostly looked at their mental health, which includes mood disorders, suicidal and non-suicidal self-injury, and anxiety disorders, in addition to their sexual and reproductive health [1]. However, it is noted that there is a dearth of research on the risk factors for such mental health problems or non-infectious reproductive health concerns [1].

For the MSM population, of the health risks that receive academic attention, HIV-related risks top the list. This concern is founded on the fact that MSM account for a significant proportion of cases in HIV and other sexually transmitted infections (STIs) clinics, as shown to be the case in Indonesia [19], Brazil [20], North America [21], Western Europe, Australia [22], and overall in most low-, middle-, and upper-income countries [23]. One study among the MSM in New York City estimates that this group has a 140-fold higher risk for newly diagnosed HIV and syphilis compared with heterosexual men [24]. Indeed, according to a comprehensive review of available data for HIV prevalence, incidence, risk factors, and the molecular epidemiology of HIV in MSM from 2007 to 2011, not only does receptive anal intercourse have a crucial role in accounting for the disproportionate disease burden in MSM, higher rates of dual-variant and multiple-variant HIV infection can also be seen in MSM than in heterosexual people in the same populations [23]. A study in Kuala Lumpur, Malaysia, notes that individuals who perceive themselves to be at high risk of HIV infection have a significantly high likelihood of testing HIV positive [14]. As such, most studies on the topic look for the causes of HIV transmission and barriers to HIV prevention. Cases in which these individuals are tested HIV positive are often associated with “multiple sex partners, multiple types of sex partners, alcohol use before intercourse, unprotected sex beyond 6 months, and inconsistent condom use during anal sex” [14]. The risky sexual practice is linked to a faster spread of HIV, according to a qualitative study among the MSM in rural southern China, which suggests that HIV prevention programs pay attention to the dual identities of many MSM who are reluctant to “come out” in fear of social stigma [2]. Another study in Chennai, south India, surveys 62 MSM outreach workers from three non-governmental organizations to find that only half of them have been tested for HIV while the majority would rather not know they had HIV until they were sick [25]. The authors then call for structural interventions in the healthcare settings and in the community.

Similar to the MSM group, clinical experience and research have invariably shown transgender people to be vulnerable to HIV infection and other STIs [26,27]. A meta-analysis of studies on HIV infection burdens in over 11,000 transgender women worldwide finds the population rate of HIV infection was as high as 19.1% [27]. More notably, the probability for testing positive to HIV in transgender women compared with all adults of reproductive age across the 15 surveyed countries was 48.8% [27]. As with the MSM population, transgenders face a high burden for HIV infection because of their risky sexual behaviors, such as inconsistent condom use and needle-sharing for hormone injections [28].

### 1.2. Research on MSM and Transgenders in Vietnam

MSM and transgender people in Vietnam face the same unfortunate plight as their international peers—under-represented, stigmatized, at high risks of HIV/STIs, and lacking access to proper healthcare [29,30,31,32,33,34,35,36,37]. The increased HIV infection among the MSM in Vietnam, including the male sex workers, is attributed to the high numbers of sexual partners, high rates of unsafe sex, and inconsistent condom use [32,38,39,40]. Additionally, the syndemic condition of psychosocial factors, such as depression and substance use, may disproportionately burden male sex workers and increase their HIV risk [41,42]. A closer look at the data shows that the HIV prevalence rate among this group has risen so rapidly, from 3.7% initially to 16% in 2017, that it is considered at the highest risk of HIV transmission in the country [29]. In particular, the rate of HIV transmission among MSM nearly doubled from 9.4% in 2006 to 17.4% in 2009 in the capital city of Hanoi, while that in the southern metropolis of Ho Chi Minh City rose even faster—from 5.3% to 16.7% during the same period [43]. According to estimations of the Vietnam authorities of HIV/acquired immunodeficiency syndrome (AIDS) control, by 2015, while the national HIV prevalence stays as low as 0.3–0.4% and remains stable, the HIV transmission rate in HCM City already climbed to 38% [43]. Besides the HIV risk, a study has shown that one in five MSM in HCM City is afflicted with one of the three STIs: syphilis, chlamydia and gonorrhea [30].

Given such risks, clinical approaches to bringing down the epidemic among the MSM population have stressed the need to improve HIV knowledge and risk factors as well as to apply integrated preventive care that take into account substance use, mental health, and stigma [30]. Notably, one study examines the technology usage to find sexual health information online among MSM in Hanoi, concluding that technology-based interventions could help reduce HIV and other STIs and even reach different subgroups [33]. Researchers particularly called for more attention to male sex workers in big Vietnamese cities as they are often disengaged from the healthcare system, and consequently contributing to large numbers of untreated or late-treated infections and increased secondary transmission [31]. This subgroup is also found to engage frequently in high-risk sexual and drug behaviors [42], and thus, are more susceptible to mental distress [44]. However, it is important to note that there is an interaction between high stigma and the risky behaviors. The stigma of the health professionals toward these groups can lead to poor treatments, thus increasing the likelihood of the patients infecting more partners within their particular sex network. 

### 1.3. Research Rationale

The literature on the topic reveals a shortage of research on the knowledge and preparations of healthcare professionals in serving gender non-conforming patients. While the MSM and transgender individuals themselves have pointed out the inadequate attention and care from health providers, there appears to be little research on the specific misperceptions and lack of understanding of this population from the latter’s perspectives. Studies, however, have noted a shortage of care providers who are culturally competent in and knowledgeable about LGBT health [45]. Meanwhile, in the case of Vietnam, researchers have mostly examined physicians’ knowledge, attitudes, and practices about HIV/AIDS prevention and treatment [34,35,46,47,48]. In particular, Platten et al. [48] point out that medical students in Hanoi lack knowledge about the categories of HIV-related basic sciences, prevention, and care and treatment. As can be seen, patients tested positive for HIV, both male and female, are already facing poor treatment and stigma [34,35,49,50], people living with HIV who belong to sexual minorities and gender non-conforming groups are undoubtedly suffering worse. The widespread and deep-rooted stigma likely stems from misinformation and social or personal biases, which have been reinforced through the conventional gender binary upbringing [18]. On these grounds, it is important to be aware of this gap of knowledge in order to devise proper solutions and medical services for these groups as well as the gender incongruence population at large. This study contributes to the small body of research on transgender people’s health care in Vietnam [29,30,31,32,33,34,35,36,37] by explaining the shortcomings of Vietnamese healthcare professionals in this matter. It invites other low- and middle-income countries to join the discussion on better ways to respond to the MSM and transgender communities.

Figure 1 outlines the basic clinical process for recognizing and meeting the needs of the MSM population. Given that misunderstanding regularly happens among health workers when dealing with this group, the first step is to correctly identify the status of a man who have sex with man. Research has confirmed that healthcare providers knowledgeable of their patients’ sexual orientation and gender identity can lead to improved access, quality of care, and outcomes [51]. Indeed non-disclosure of MSM status to healthcare providers in Vietnam has been shown to cause suboptimal screening for HIV and sexually transmitted infections (STIs) and late detection of secondary transmission [29]. Once there has been correct recognition, health professionals can begin laying out the needed mental and physical care for the patients, which should be followed by periodic health check-ups and HIV/STI testing.

Based on this framework, this study, with the hope to alleviate the risk of policy failures [52,53], delves into the specificity of the first step, such as the common misunderstanding or lack of understanding among Vietnamese health workers when serving MSM and transgender patients. 

## 2. Materials and Methods

### 2.1. Study Design and Settings

This study was conducted by the Letter of Approval #ISDS-03-2016 issued by the Institute of Social Development Studies (ISDS), per the request from the Vietnamese Center for Supporting Community Development Initiatives (SCDI). This was within the component (CD26) aimed at supporting men who have sex with men and transgenders of the PITCH project (Partnership to Inspire, Transform and Connect the HIV response). PITCH, funded by the Dutch government, is a strategic partnership between Aidsfonds, the International HIV/AIDS Alliance, and the Dutch Ministry of Foreign Affairs [54]. 

The research team applied both quantitative and qualitative methods in surveying health workers at public and private hospitals/clinics in Vietnam’s two biggest cities, namely Hanoi and Ho Chi Minh City, concerning their knowledge, attitudes, and behaviors towards MSM and transgender patients. The survey was based on the convenience sampling method and these hospitals and clinics are chosen on the basis they have experienced working with MSM/transgenders. It is worth noting that Vietnamese dermatology clinics/hospitals use the same system as the European system of combining training and treatment for dermato-venereology. This means that patient inquiries on venereology diagnosis and treatment can be answered by doctors at this type of hospital.

The respondents include physicians working in clinical settings and meeting patients on a daily basis as well as health professionals serving at leadership roles. In particular, the team carried out (i) a survey among 150 health workers that covered 61 questions and (ii) 12 in-depth interviews based on an internal interview guideline that has integrated relevant materials and opinions of experts in the field. The response rate for the survey was 100%. 

All questions are centered around three main topics: (i) the general knowledge of healthcare workers about MSM and transgenders; (ii) their knowledge about the sexual reproductive health and HIV/AIDS risks of MSM and transgenders; and (iii) their attitudes and behaviors towards MSM and transgenders.

### 2.2. Participants

In total, the research surveyed 150 respondents and conducted 12 in-depth interviews, which were divided equally by the two surveyed locations. No respondents had to disclose their names or other personal information for confidentiality purposes. Table 1 describes the general information on the survey respondents.

Although the percentage of female respondents to the survey is three times that of males, the majority of these female respondents work in administrative roles, whereas most male respondents hold management or specialist positions. Figure 2 illustrates this observation.

Data collected from the survey were input and analyzed using the SPSS software (version 24.0) while information collected from the in-depth interviews were recorded, transcribed and compiled using the NVivo software (version 1.0.226.0). Table 2 summarizes the background information on 12 participants to the in-depth interviews.

As Table 2 shows, the research team only conducted in-depth interviews with health workers in the specialist departments, such as doctors or nurses, and those holding management or leadership roles. Due to the short amount of time and the limited sample size, the in-depth interviews selected only participants who have significant experience or more experience working with MSM and transgenders.

## 3. Results and interpretations

### 3.1. General Knowledge about MSM and Transgenders

#### 3.1.1. Concerning MSM

In order to assess the general knowledge of the health workers on MSM and TG, the research team listed 21 common perceptions of the two groups, including the misconceptions, and requested the participants to select whether they agree with each perception or not. Table 3 summarizes the results.

Table 3 shows the majority of respondents agree with the statement that MSM have a typical male’s genitalia and they can have sex with both males and females (81.6% and 74.8%). However, up to 45% of the respondents still believe that MSM only have sex with males. This means it is necessary to provide health workers with knowledge about the diversity in sexual orientations, sexual identity and sexual behaviors of MSM. Such adequate knowledge is crucial for providing more suitable services for this at-risk population.

There are still many respondents who agree with the incorrect statements about MSM or even express a degree of discrimination with MSM. Over half of the participants think that MSM are those with sex hormone disorder, one-third think MSM are androgynous, and nearly one-third perceives MSM are those with effeminate behaviors that are not suitable for their gender. More remarkably, there is a 17% of the health workers in this study still think MSM is a curable mental problem. This is an outdated viewpoint and must be discarded to reduce discrimination and stigma towards MSM and to enhance the effectiveness of medical treatment for this group.

Box 1Health workers’ understanding of MSM identity.“MSM people also belong to the third sex, meaning they are men but they have sex with people of the same sex and they don’t like women, and they don’t like heterosexual relationship as well as a marriage life of normal people.” *(Female, Management/Leadership, MA, General Hospital, Hanoi)*“Q: Which characteristics make you think they are MSM?”A: Honestly speaking, there are many sensitive matters. For example, their attitude when talking, their “polished” way of dressing, or their attention to beauty, or their effeminate manner.” *(Male, Department Head, MA, General Hospital, Ho Chi Minh City)*
“In my opinion, how to identify their gender, they (MSM) are not aligned with the biological structure […]. For example, they are male. But in their mind, they are always interested in males rather than females. If they behave according to their sex, they would be interested in females.” *(Female, Head Nurse, Andrology Center, Hanoi)*

Many medical staff share that they identify which patients are MSM based on appearances. For example, if the clothes or behaviors are similar to a typical female although the individuals are male, staff would identify them as MSM. While this crude “method of identification” can be useful when the patients are transgenders, a large number of MSM do not dress or act in this stereotypical description. Here, it is important to remark that the diversity of the MSM community has not been fully recognized.

In particular, Figure 3a shows that the number of health workers who are female and have lower educational level (college or below) holding incorrect understanding of MSM is higher than that of those who are male and have higher educational level. This disparity is also reflected when respondents are categorized by different levels of work position. In Figure 3b, health workers in administrative roles have less knowledge about MSM than those in either specialist or management/leadership positions.

When categorized by medical facilities and cities, as shown in Table 4, there are more health workers in Hanoi who misperceive the identity of MSM than those in Ho Chi Minh City (HCMC). Andrology clinics and private clinics have the highest percentage of people who agree with incorrect statements about MSM.

#### 3.1.2. Concerning Transgenders

Similar to the section on MSM, the following questionnaire presents 12 statements about transgenders for the participants to select whether they agree with each sentence or not. Table 5 summarizes the results.

The results in Table 5 can be viewed with some measure of optimism. Most respondents agree with the correct views on transgenders and only 2% of the health workers think transgenders have a mental problem and can be cured. Nonetheless, it is alarming that 50% of the respondents consider transgenders to be people with a sex hormone disorder. In fact, this problem arises only after these individuals undergo hormonal therapy to change their sex.

When categorized by education and work positions, the majority of health workers who misperceive transgenders belong to the group with low educational level and administrative-related departments, whereas there are fewer specialist physicians or those holding management/leadership roles with higher educational level agreeing with incorrect statements about transgenders. Figure 4 illustrates clearly the result that most people who hold incorrect views about transgenders are those who have lower educational levels or work in administrative positions at medical facilities (Table 6).

Box 2 presents some statements made by health workers regarding their understanding of transgender people.

Box 2Health workers’ understanding of transgender identity.“In my opinion, for those who want to be transgenders, their biology is different from their psychology. For example, if a person is male, and he has male genitalia, but all his thoughts, expressions, and mentality are of females.” *(Male, Doctor, Dermatology Hospital, Ho Chi Minh City)*“For example, for those who change from female to male, it means their body is of a female, but in their thoughts, they always think they are male, they might dress like a man, their clothes and personalities are more masculine, they always desire their body to be a male body and they have male desires.” *(Female, Head Nurse, Dermatology Hospital, Hanoi)*“The transgenders are those who change from one gender to another, for example, those who were born as male and want to become female.” *(Female, Management/Leadership, General Hospital, Hanoi)*“The transgenders are homosexuals so they must change their sex to make it more suitable, this is what I think.” *(Female, Doctor, General Hospital, Hanoi)*

It is noteworthy that when speaking of transgenders, some health workers only think of the transgenders who have medical interventions such as hormonal therapy or operation. They do not know that there are many transgenders who do not have this demand or desire to change their body, or who have such wish but their conditions are not favorable to carry out this change.

### 3.2. General Knowledge about Sexual Reproductive Health and HIV/AIDS Risks of MSM and Transgenders

#### 3.2.1. Concerning MSM

The study finds a high percentage of health workers having correct understanding of the variety of sexual intercourses practiced by MSM, such as 97.3% agreeing with oral sex, 83.6% with genital sex, and 95.2% with anal sex. The number of male health workers who hold these correct understanding is higher than that of females (Figure 5). The same trend is seen in the group of those with university education or higher versus those with college or lower (Figure 5). No significant differences are seen between the two cities or when categorized by work positions or medical facilities.

When asked about the sexual reproductive health (SRH) issues that MSM often face, the majority of respondents agreed with all 25 examples listed in the questionnaire. However, respondents appeared ambivalent about the following five issues as less than half the survey sample agreed that these are the common SRH issues for MSM: premature ejaculation (49.7%), groin lump/ swollen lymph gland (49%), erectile dysfunction (47%), testicular cancer (37.6%), and prostate cancer (34.9%). On the other hand, most health workers agreed that the following five SRH issues are common in MSM: 99.3% (99.3%), gonorrhea (98%), genital warts (96.6%), HIV (94.6%), and anal warts (91.3%).

#### 3.2.2. Concerning Transgenders

Health workers in the survey mostly agreed on the two main SRH issues facing transgender people, namely anal warts (87.7%) and anal fissure (81.5%). The other two issues that respondents appeared unclear about are erectile dysfunction (44.5%) and anus/rectum leakage (54.8%). Similar to the knowledge about MSM, there are more female health workers lacking knowledge about these issues than males (Figure 6a). Additionally, staff at dermatology hospitals lack understanding of SRH issues facing transgenders when compared with peers at general hospitals and, especially, andrology clinics and private clinics (Figure 6b). No differences are seen in the two cities or when categorized by work positions and educational levels.

Besides the information on SRH, participants were also asked whether they know about the behaviors that are health-related or the health issues typically faced by transgenders such as hormone self-injection, unsafe multiple surgeries to change the body, unsafe sex, self-treatment, overuse of hormone therapy, endocrine disorder due to using hormones, infections/complications due to sex reassignment surgery. The majority of health workers know about these risky behaviors of transgenders; however, the two least known problems are the self-injection of hormones and self-treatment (only 75.7% and 67.1% know about these problems, respectively). The health workers seem to have a good level of general knowledge on the health behaviors and problems typically faced by transgenders. The respondents understand the complications and high risks when the transgender patients try to change their bodies. 

The majority of the participants said they do not meet with transgenders. Those who work directly with transgenders said that the transgenders often come to their medical facilities for general health-checkups or for common illness. There is no health worker who had handled the health problems typically faced by transgenders. In fact, no facilities provide any service that cater to the needs of transgenders. The reason is that until now, the regulation and guidelines related to sex reassignment and hormone therapy for transgenders have not been recognized in Vietnam. 

Box 3Health workers’ understanding of the sexual reproductive health issues facing transgenders.“Actually, only a few the transgenders I met have STDs, perhaps, because when they come to have a diagnosis, they might have a disease but they are not in my prognosis. The male homosexuals I met usually have genital warts.” *(Female, Management/Leadership, 18 years of experience, Dermatology Hospital, Hanoi)*“They (transgenders) are diagnosed with the general illnesses such as influenzas, runny nose, pneumonia, they never ask about hormones because they know we are not specialists.” *(Male, Doctor, 10 years of experience, Private clinic, Ho Chi Minh City)*

With regard to the HIV risks, almost all the participants (87.3%) recognize MSM and transgenders as the community with higher risks of HIV infection compared to other populations—this is a correct perception according to the literature [32,38,39,40]. 

As Figure 7a shows, there is a gap of understanding between the health workers in Hanoi and Ho Chi Minh City (94.7% vs. 79.7%). This disparity is also observed in the different medical facilities, per Figure 7b, such that the district-level general hospitals seem to either underestimate the HIV risks or be completely unaware of such risks faced by this sexual minority group. By comparison, private clinics and municipal dermatology hospitals appear to be more knowledgeable of the transmission risks. When work positions are taken into account, as Figure 7c shows, there is also a considerable gap of understanding, with more people holding management/leadership and specialist roles being informed about the transgenders’ HIV risks than those in administrative positions.

Most respondents (81.5%) attributed why MSM and transgenders are the population at the highest risk of HIV infection to the population’s tendency to have anal sex, as the anus is a very vulnerable body part. The second popular reason is MSM and transgenders have many sexual partners (79.2%), next is they do not use protection correctly when having sex (78.5%). Three reasons that are least chosen are: they have high frequency of sexual intercourse (35.4%), they use lubricant incorrectly (28.5%), and they share needles (26.9%).

This information should be better explained to the health workers, especially as it is known that transgenders in Vietnam often practice hormone self-injection and share needles to lower cost [37]. This is a serious matter for transgenders but is not known widely by the health workers. 

Box 4A health professional in leadership role discusses the HIV risks facing MSM and transgenders.“In fact, HIV-infection is caused through having anal sex, this type of sexual intercourse increases the risk of infection much higher than the traditional vaginal sex. This is because there would be more scratches.” *(Male, Leadership, General Hospital, Ho Chi Minh City)*

### 3.3. The Attitudes and Behaviors of Health Professionals towards MSM and Transgenders

To assess the attitude of the health workers toward MSM and transgenders, this study uses a questionnaire of 11 statements, in which some are discriminatory. In addition, we also ask about the workers’ level of concern about HIV-infection when providing services for MSM and transgenders and ask about how they use gender pronouns. 

It is notable that a high percentage of respondents agree with the discriminatory statements toward MSM and transgenders, even the harsh ones; 18.7% of the respondents think that MSM and transgenders just imitate the Western way of life; 14% agree with the statement that transgenders who want to change their sex cause troubles for their family and society; 12.7% say if they get to choose, they want nothing to do with MSM or transgenders including providing services to them. Over 7% state that MSM and transgenders are perverts and the use of hormones or sex reassignment surgery are unnatural and perverse. Finally, 3.3% believe MSM and transgenders are promiscuous and immoral.

As shown in Figure 8, the striking result in 4 out of 5 discriminatory statements is the percentage of the health workers who claim to interact frequently with MSM and transgenders is higher than those who seldom or never work with these populations. 

Regarding the use of gender pronouns for MSM and transgenders, 42% of the health workers in this study state they do ask these groups how they want to be addressed. However, many workers in the in-depth interviews reveal they do not act like that, rather, they address either spontaneously by their own will or based on their identification documents. There are cases of misunderstanding that lead to the MSM and transgenders feeling uncomfortable when they are addressed wrongly or feel the health workers are not sensitive (See Box 5). 

Box 5Health workers discuss the issue of gender pronouns when dealing with MSM and transgenders.“Honestly speaking, I have never asked them how they’d like to be addressed?” *(Male, Leadership, Andrology Clinic, Hanoi)*
 “A: To address the patients, if he is older than me, I call him “anh”; if he is younger, I call him “em.” There is nothing special. Q: Have you ever asked them how they’d like to be addressed? A: No, I haven’t.” *(Female, Nurse, Dermatology Hospital, Hanoi)* “The nurses and the receptionists sometimes see a male name in the identification cards or social insurance cards, but the person seems to be female. They ask whether she uses the cards from someone else. There are many cases like this.” *(Male, Doctor, Private Clinic, Ho Chi Minh City)*

The results from the survey corroborate this insight. There is a higher percentage of health workers in administration who show a lack of sensitivity when interacting with MSM and transgenders, with 36%. While for people in leadership/management position or specialists, there is a higher percentage who understand asking MSM and transgenders how they’d like to be addressed is something one ought to do (50% and 44% respectively), as shown in Figure 9.

This study also evaluates the knowledge and attitude of health workers on the risk of HIV-infection when providing services to MSM and transgenders. One striking result is that 18% admit they are worried about being infected with HIV when in contact with clothes, blankets or beds of the MSM and transgenders. Moreover, up to 63% of the respondents worry about getting HIV-infected when treating the wounds or taking blood sample from these two groups. These results indicate many health workers do not have adequate knowledge about HIV and the standard prevention measure, on the one hand. They also show that MSM and transgenders are facing “double discrimination” when they are equated with HIV patients, on the other hand. 

## 4. Discussion

### 4.1. Limitations

This study employs descriptive statistics and in-depth interviews to give a comprehensive picture of the general knowledge and attitude of Vietnamese health professionals toward MSM and transgenders. Thus, the study can benefit from further correlational analyses of certain (mis)perceptions and different demographic and socioeconomic factors, such as in [55,56,57]. Given that the sample size is 150 participants, in which various demographic and social characteristics are represented, this can be done. Moreover, with the convenience sampling method, the results presented in this paper should not be generalized to all healthcare facilities across Vietnam. As the study chose the hospitals/clinics in cosmopolitan areas in Vietnam and those who have worked with the MSM and transgenders before, one can reason that the misperceptions about these minorities might be much more common among the general population of healthcare workers. As noted in the methods section, the interviewees are limited to only those with significant experience or more experience working with MSM and transgenders, and the results must be interpreted with caution.

### 4.2. Implications

First, the results highlight a clear lack of understanding among Vietnamese health professionals when it comes to identifying MSM and transgenders. This misperception is observed in workers of both sexes, in both cities, at various kinds of medical facilities, at different work positions and educational levels. Health workers are particularly unaware of the diversity in sexual orientations, sexual identity and sexual behaviors of this gender non-conforming group. As such, they often end up identifying or addressing their clients based on stereotypical appearances. A “mythology” about MSM and transgenders can develop through a combination of a number of factors: stereotypical and self-fulfilling observations of clinicians toward minorities; the lack of continual professional and in-school training; and finally a biased admission rate of being MSM as people who have sex with both men and women are not likely to admit they are MSM. 

Notably, there is also significant confusion among health staff regarding the differences between MSM and transgenders. Among the reasons for this conceptual confusion are perhaps the cultural factors; the Vietnamese terms used for the MSM, transgenders, and homosexuals are not straightforward translation from the English terms. For example, MSM in Vietnamese are usually referred to as “male homosexuals” (*đồng tính nam*) or “male who have homosexual relationships” (*nam có quan hệ đồng tính*), while transgender in Vietnamese is referred to as “a person who changes his/her sex” (*người chuyển giới*). Hence, although many participants in this study possess a good level of general knowledge about MSM and transgenders, they remain confused when being asked about more in-depth issues, especially the diversity of this community. Here, it is paramount to understand the cultural and philosophical underpinnings of attitudes toward sexual minorities [58,59] when supplying accurate definitions and knowledge about MSM and transgender to health workers.

Second, when various demographic and socioeconomic factors are taken into account, the findings show that the groups which appear less informed about the health risks and related issues facing MSM and transgenders are often: female, having lower educational levels, working in administrative roles or at district-level hospitals. There are two caveats here: the number of female respondents is three times as much as that of males in this study, and there is a large number of female respondents holding administrative positions. The only time a difference is observed between the two cities is when respondents were asked about their knowledge of the HIV transmission risks faced by this sexual minority group. Here, it is unclear why there are more physicians in Ho Chi Minh City being aware of the high HIV infection risks than in Hanoi. The implication, then, is clear: training courses for health workers that lump together health workers in all demographic groups and departmental positions could end up being ineffective. Given this gap of knowledge, the courses on general knowledge about MSM and transgenders as well as their sexual reproductive health/HIV risks should take into account these findings to generate suitable programs for each city and groups of staff.

Last but not least, it is crucial to recognize the importance of finding an institutional fix for the high prevalence of misunderstanding toward MSM and transgenders among the clinicians demonstrated in this study. Adding updated information about MSM and transgenders into the formal training curriculum at schools as well as offering frequent in-job training are among the cost-effective and easy ways to substantially improve the situation. 

## Figures and Tables

**Figure 1 jcm-08-00105-f001:**
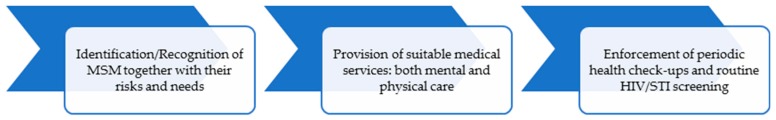
The clinical process for meeting the needs of men who have sex with men (MSM). HIV, human immunodeficiency virus; STI, sexually transmitted infection.

**Figure 2 jcm-08-00105-f002:**
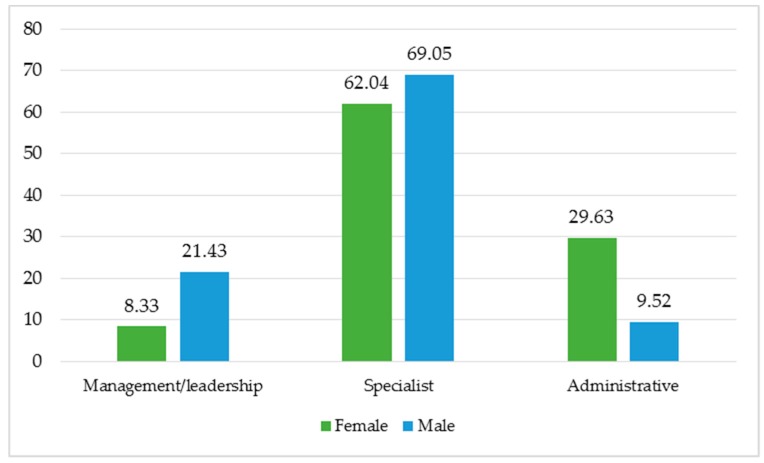
The percentage of respondents to the survey categorized by sex and work positions.

**Figure 3 jcm-08-00105-f003:**
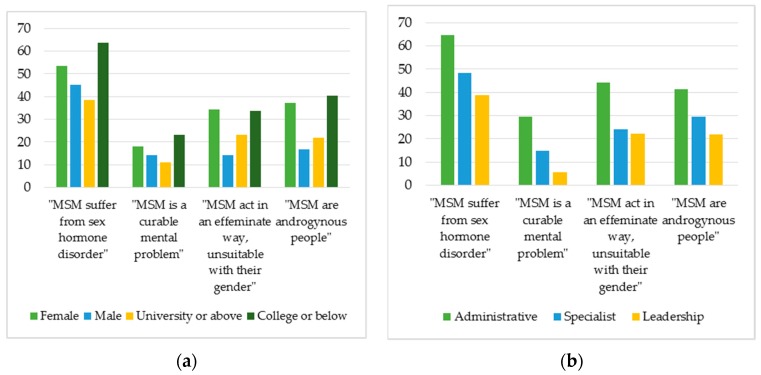
The percentage (%) of respondents agreeing with incorrect statements about MSM as categorized by: (**a**) sex and educational levels; and (**b**) work positions.

**Figure 4 jcm-08-00105-f004:**
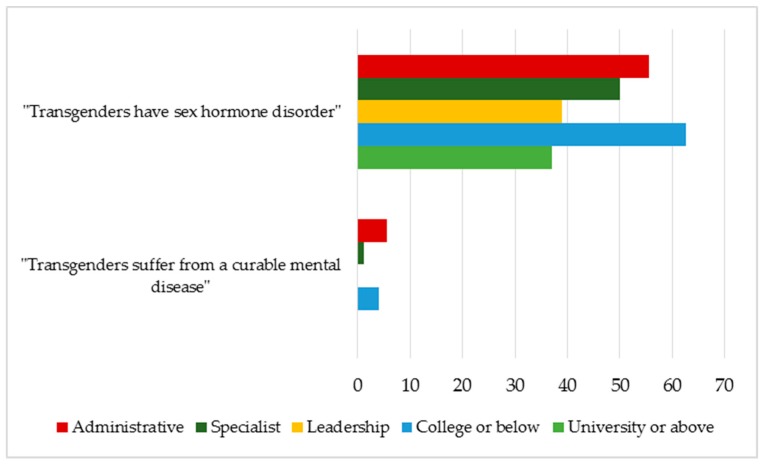
The percentage of respondents agreeing with the incorrect statements about transgenders as categorized by educational levels and work positions.

**Figure 5 jcm-08-00105-f005:**
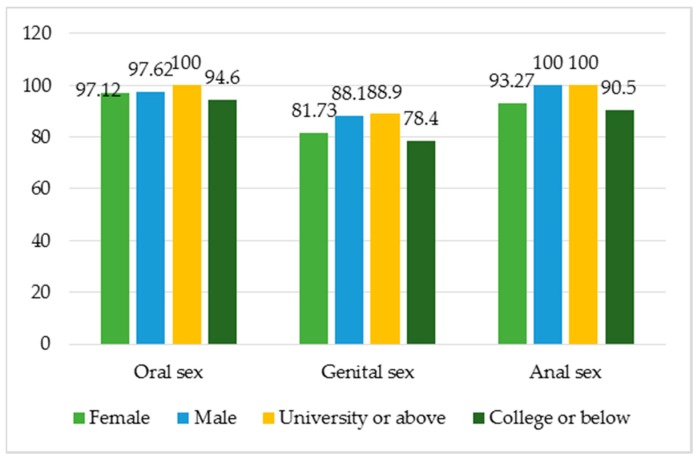
The percentage of respondents agreeing with the different ways of sexual intercourse practiced by MSM as categorized by sex and educational levels.

**Figure 6 jcm-08-00105-f006:**
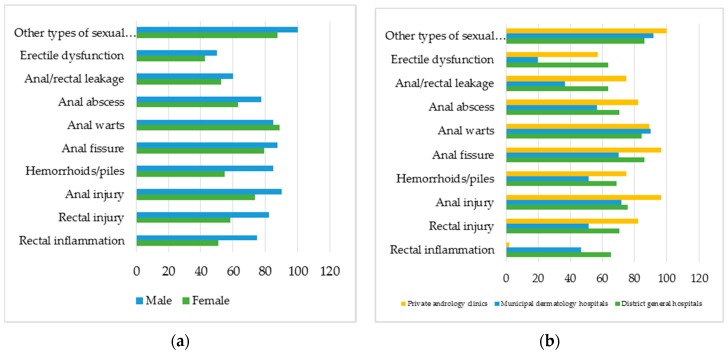
The percentage of respondents agreeing with common SRH issues facing transgenders as categorized by: (**a**) sex, and (**b**) medical facilities.

**Figure 7 jcm-08-00105-f007:**
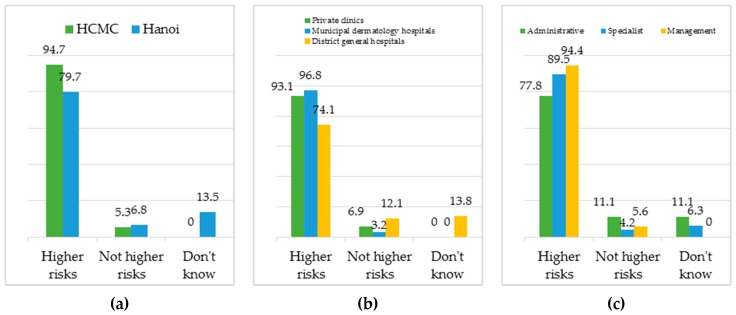
The percentage (%) of respondents knowledgeable about the HIV transmission risks faced by MSM and transgenders as categorized by: (**a**) cities, (**b**) medical facilities, and (**c**) work positions.

**Figure 8 jcm-08-00105-f008:**
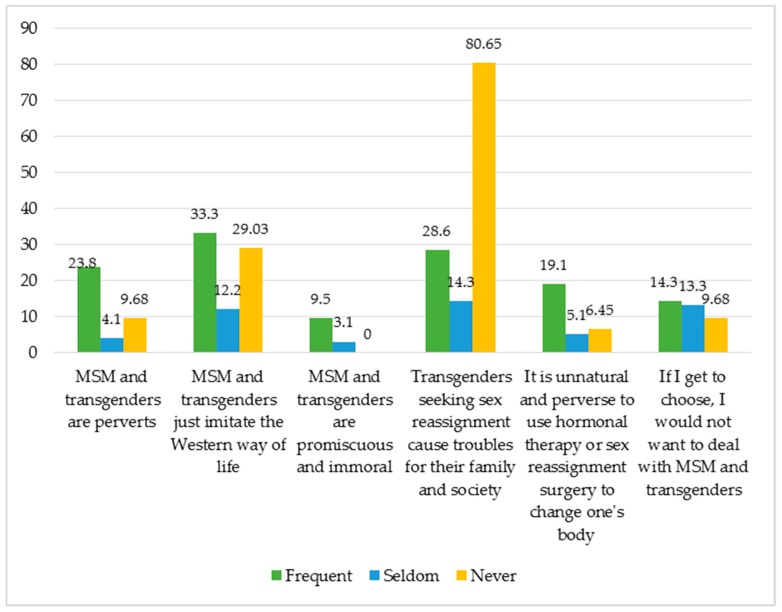
The percentage of respondents agreeing with the discriminatory statements about MSM and transgenders as categorized by their frequency of interaction with MSM and transgenders.

**Figure 9 jcm-08-00105-f009:**
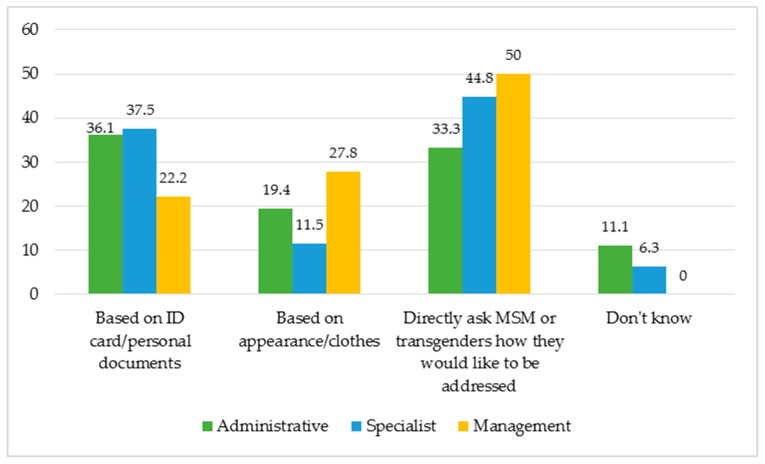
The different ways in which health workers address their clients as categorized by work positions.

**Table 1 jcm-08-00105-t001:** Background characteristics of survey respondents.

	Percentage (%)
**Sex**	
Male	28.0
Female	72.0
**Age**	
<30	38.3
30–45	47.0
>45	14.8
**Education**	
University and above	50.7
College/vocational college or lower	49.3
**Medical facilities**	
Public hospitals	38.7
Dermatology hospitals	41.3
Andrology clinics and private clinics	20.0
**Work positions**	
Management/leadership	12.0
Specialist	64.0
Administrative	24.0
**Years of experience**	
<5 years	39.6
5–10 years	26.2
>10 years	34.2
**Total**	100

**Table 2 jcm-08-00105-t002:** Background characteristics of in-depth interviewees.

	Number
**Sex**	
Male	6
Female	6
**Medical facilities**	
General hospitals	4
Dermatology hospitals	4
Andrology clinics and private clinics	4
**Work positions**	
Management/leadership	4
Specialist	8
**Years of experience**	
<5 years	1
5–10 years	3
>10 years	8
**Total**	12

**Table 3 jcm-08-00105-t003:** The percentage (%) of respondents who agree with the statements about MSM.

	Statements	Percentage (%)
1	MSM are males who only have sex with males	45.6
2	MSM can have sex with both males and females	74.8
3	MSM are those with sex hormone disorder	51.0
4	MSM are those with curable mental problem	17.0
5	MSM are those with effeminate behaviors that are not suitable for their sex	28.6
6	MSM are androgynous	31.3
7	MSM have typical male genitalia	81.6
8	MSM may have typical female genitalia	19.1
9	MSM may not have any genitalia characteristics of either sex	29.9

**Table 4 jcm-08-00105-t004:** The percentage (%) of respondents agreeing with the incorrect statements about MSM as categorized by medical facilities and cities (Hanoi and HCMC).

Incorrect Statements	General Hospitals		Dermatology Hospitals		Andrology Clinics and Private Clinics	
	Hanoi	HCMC	Hanoi	HCMC	Hanoi	HCMC
MSM are those with sex hormone disorders	60.0	60.0	65.6	30.0	69.2	56.3
MSM are those with a curable mental problem	10.0	10.0	18.8	6.7	30.8	6.3
MSM are those with effeminate behaviors that are not suitable for their sex	30.0	30.0	43.8	6.7	53.9	12.5
MSM are androgynous	33.3	33.3	53.1	3.3	61.5	31.3

**Table 5 jcm-08-00105-t005:** The percentage (%) of respondents who agree with the statements about transgenders.

	Statements	Percentage (%)
1	Female transgenders are those who were born as males but feel and express themselves as females	85.8
2	Male transgenders are those who were born as females but feel and express themselves as males	81.1
3	Female transgenders are those who were born as male but use medical intervention to have a female body	82.4
4	Male transgenders are those who were born as male but use medical intervention to have a male body	74.3
5	Transgenders have a curable mental problem	2.0
6	Transgender are those with sex hormone disorder	50.0
7	Transgenders may have typical male genitalia	68.9
8	Transgenders may have typical female genitalia	72.3
9	Transgenders may have genitalia atypical of both sexes	34.5
10	Transgenders may not want to alter their body but want to be acknowledged as members of the sex different from their innate sex	62.8
11	A male transgender may have sex with both males and females	74.3
12	A female transgender may have sex with both males and females	75.7

**Table 6 jcm-08-00105-t006:** The percentage (%) of respondents agreeing with incorrect statements about MSM as categorized by medical facilities and cities

Incorrect Statements	General Hospitals		Dermatology Hospitals		Andrology Clinics and Private Clinics	
	Hanoi	HCMC	Hanoi	HCMC	Hanoi	HCMC
Transgenders are those with a curable mental problem	3.3	0.0	0.0	0.0	15.4	0.0
Transgenders are those with sex hormone disorders	63.3	38.5	46.9	40.0	69.2	52.9

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
