# Peer review of "Righting the Misperceptions of Men Having Sex with Men: A Pre-Requisite for Protecting and Understanding Gender Incongruence in Vietnam"

_jcm, 2019, doi:10.3390/jcm8010105_

Round 1
Reviewer 1 Report
This is an interesting and useful paper, with implications well beyond VietNam. The authors need to be careful not to state that HIV/STIs have become prevalent among MSM. That is different to saying that for STI prevalence, MSM account for a significant proportion of cases in HIV/STI clinics (which is correct). One would need a random sample of MSM to determine that there was a high prevalence in MSM (and not just those who identify as such to the health service). It is not clear whether MSM, or MSM and transgendered individuals at highest risk (including some sex workers), selectively attend certain clinics, which could also bias findings. It is dangerous to attribute high levels of HIV/STI to a whole minority population just because there is a subpopulation which has high HIV/STI rates in particular clinic samples. A high representation of MSM in HIV/STI diagnoses in clinics does not necessarily mean the converse, that HIV/STIs are high or increasingly prevalent in MSM. It’s wiser to cite population rates than to make generalizations.
It is also important to be aware that there is an interaction between high stigma and risky behaviors. Higher depression, other mental distress and internalized homonegativity/homophobia are related to increased risk, and so a simplistic association that it’s just the behavior that causes HIV/STI rates to be higher in some sexual minority groups seems like “victim-blaming”. There are complex interactions between being a minority and higher stigma and depression and higher risk behaviors. For example, if MSM and transgender people are stigmatized or given poor treatment in some clinics, then that delay or lack of treatment will increase the number of partners they likely have while they are infectious within their particular sexual network. This is certainly acknowledged at the bottom of p3 (ref. 31).
Further, data also confirm that a relatively small sexual network (e.g., transgender, MSM and men who are their partners) will magnify infection rates as a function of the limited number of possible partners and the higher chance of meeting with an infected partner. I make this point not because I disagree with the data in this paper, which I don’t, but because I think that the conclusion that MSM = high HIV/STIs is dangerous and simplistic and ignores good social science and policy evidence that stigma and discrimination based on minority status, and smaller sexual networks, can increase risk and infection.
It would be interesting to know what proportion of MSM admit to their being MSM in clinics. There is evidence from other countries that it is often only those with anal infections or lesions who admit, and this may create a patient stereotype in clinicians. Nor is it clear whether MSM who have sex with both men and women admit to being MSM – thus missing perhaps the largest group of MSM in a society. Some comments on how a “mythology” about MSM can develop due to biased rates of admitting to MSM contact (along with a lack of education on sexual health and behavior in medical or nursing schools) should be made. These data suggest that a combination of absence of professional and continuing education training, and the development of a “mythology” based on self-fulfilling observations in a clinical setting, is a factor in these data.
I think that the authors have to some extent missed the main message from these data: that absence of adequate (or any) information on sexual health and behavior as part of medical and nursing school training is a primary cause of the important misinformation that seems to occur with regard to clinic professionals. This is compounded by lack of continuing (e.g. clinic presentations and lectures) professional education – which is easier and quicker to correct than providing comprehensive medical and nursing school curricula. Whatever happened to in-job education (lectures, workshops) in the clinics surveyed? It deserves more than a sentence at the end of the paper as a relatively cheap and easy way of correcting some of the sources of misinformation. It is unfortunately too easy to “victim blame” health professionals and staff for their lack of information where the medical and nursing school curricula, and clinic/continuing education for staff on sexual health have been absent, inadequate or neglected.
Finally, reference is made to dermatology clinics/hospitals and I wonder if what is being referred to is actually venereology or STI clinics. The European system of combining training and treatment for dermato-venereology will not be understood in places such as North America and the British system where in fact dermatology is a separate discipline and training from venereology/STI specialization. Clarification here would be helpful.
This manuscript has scientific and professional resonance well beyond VietNam and addresses an important issue.
Author Response
Dear Sir/Madam,
We would like to take this opportunity to express our deep appreciation for providing very helpful and detailed criticisms for our manuscript. Your meticulous input has helped us improve our manuscript significantly. In this revision, we have addressed your concerns in. Please notice that in the revised paper, the parts that are highlighted in yellow is for correction on the old text, the parts highlighted in green is written anew. Below are our answers to your comments (in italic). Also, the line numbers in the bold text refer to the revised paper.
This is an interesting and useful paper, with implications well beyond Vietnam. The authors need to be careful not to state that HIV/STIs have become prevalent among MSM. That is different to saying that for STI prevalence, MSM account for a significant proportion of cases in HIV/STI clinics (which is correct). One would need a random sample of MSM to determine that there was a high prevalence in MSM (and not just those who identify as such to the health service). It is not clear whether MSM, or MSM and transgendered individuals at highest risk (including some sex workers), selectively attend certain clinics, which could also bias findings. It is dangerous to attribute high levels of HIV/STI to a whole minority population just because there is a subpopulation which has high HIV/STI rates in particular clinic samples. A high representation of MSM in HIV/STI diagnoses in clinics does not necessarily mean the converse, that HIV/STIs are high or increasingly prevalent in MSM. It’s wiser to cite population rates than to make generalizations.
Thank you for your very detailed comments. We agree with your comment on the way we phrase the issue and have changed the wording accordingly.
Line 90-91: “This concern is founded on the fact that MSM account for a significant proportion of cases in HIV/STIs clinics…”
Line 114-116: “A meta-analysis of studies on HIV infection burdens in over 11,000 transgender women worldwide finds the population’s rate of HIV infection was as high as 19.1% [27].”
It is also important to be aware that there is an interaction between high stigma and risky behaviors. Higher depression, other mental distress and internalized homonegativity/homophobia are related to increased risk, and so a simplistic association that it’s just the behavior that causes HIV/STI rates to be higher in some sexual minority groups seems like “victim-blaming”. There are complex interactions between being a minority and higher stigma and depression and higher risk behaviors. For example, if MSM and transgender people are stigmatized or given poor treatment in some clinics, then that delay or lack of treatment will increase the number of partners they likely have while they are infectious within their particular sexual network. This is certainly acknowledged at the bottom of p3 (ref. 31).
Further, data also confirm that a relatively small sexual network (e.g., transgender, MSM and men who are their partners) will magnify infection rates as a function of the limited number of possible partners and the higher chance of meeting with an infected partner. I make this point not because I disagree with the data in this paper, which I don’t, but because I think that the conclusion that MSM = high HIV/STIs is dangerous and simplistic and ignores good social science and policy evidence that stigma and discrimination based on minority status, and smaller sexual networks, can increase risk and infection.
Thank you for your insights. We have acknowledged the relationship between high stigma and risky behaviors in the main text (Line 146-149):
“However, it is important to note that there is an interaction between high stigma and the risky behaviors. The stigma of the health professionals toward these groups can lead to poor treatments, thus increasing the likelihood of the patients infecting more partners within their particular sex network.”
It would be interesting to know what proportion of MSM admit to their being MSM in clinics. There is evidence from other countries that it is often only those with anal infections or lesions who admit, and this may create a patient stereotype in clinicians. Nor is it clear whether MSM who have sex with both men and women admit to being MSM – thus missing perhaps the largest group of MSM in a society. Some comments on how a “mythology” about MSM can develop due to biased rates of admitting to MSM contact (along with a lack of education on sexual health and behavior in medical or nursing schools) should be made. These data suggest that a combination of absence of professional and continuing education training, and the development of a “mythology” based on self-fulfilling observations in a clinical setting, is a factor in these data.
Thank you very much for your detailed feedback. We understand your concerns and apologize that due to the focus on the knowledge and attitude of health workers, we did not have any figures regarding the proportion of MSM who admit to being MSM in clinics. However, we have added a comment on the “mythology” as follows (Line 458-462):
“A “mythology” about MSM and transgenders can develop through a combination of a number of factors: stereotypical and self-fulfilling observations of clinicians toward the minorities; the lack of continual professional and in-school training; and finally the biased admission rate of being MSM as people who have sex with both men and women are not likely to admit they are MSM.”
I think that the authors have to some extent missed the main message from these data: that absence of adequate (or any) information on sexual health and behavior as part of medical and nursing school training is a primary cause of the important misinformation that seems to occur with regard to clinic professionals. This is compounded by lack of continuing (e.g. clinic presentations and lectures) professional education – which is easier and quicker to correct than providing comprehensive medical and nursing school curricula. Whatever happened to in-job education (lectures, workshops) in the clinics surveyed? It deserves more than a sentence at the end of the paper as a relatively cheap and easy way of correcting some of the sources of misinformation. It is unfortunately too easy to “victim blame” health professionals and staff for their lack of information where the medical and nursing school curricula, and clinic/continuing education for staff on sexual health have been absent, inadequate or neglected.
Thank you for raising such an important implication for your study. We have put in the end of the Discussion section a paragraph on the importance of finding an institutional fix for the problem:
“Last but not least, it is crucial to recognize the importance of finding an institutional fix for the high prevalence of misunderstanding toward MSM and transgenders among the clinicians demonstrated in this study. Adding the updated information about MSM and transgenders into the formal training curriculum at schools as well as offering frequent in-job trainings are among the cost-effective and easy way to substantially improve the situation.”
Finally, reference is made to dermatology clinics/hospitals and I wonder if what is being referred to is actually venereology or STI clinics. The European system of combining training and treatment for dermato-venereology will not be understood in places such as North America and the British system where in fact dermatology is a separate discipline and training from venereology/STI specialization. Clarification here would be helpful. This manuscript has scientific and professional resonance well beyond VietNam and addresses an important issue.
Thank you for allowing us to clarify this important distinction. The Vietnamese dermatology clinics/hospitals use the same system as the European system of combining training and treatment for dermato-venereology. The leading Vietnamese medical institute for this is called “National Hospital of Dermatology and Venereology” in official English. Patient inquiries on venereology diagnosis and treatment can be answered by doctors at this type of hospital. We have added this clarification in the “Study design and settings” section (lines 197-206):
“The survey was based on the convenience sampling method and these hospitals and clinics are chosen on the basis they have experienced working with MSM/transgenders. It is worth noting that the Vietnamese dermatology clinics/hospitals use the same system as the European system of combining training and treatment for dermato-venereology. This means that patient inquiries on venereology diagnosis and treatment can be answered by doctors at this type of hospital
The respondents include physicians working in clinical settings and meeting patients on a daily basis as well as health professionals serving at leadership roles. In particular, the team carried out (i) a survey among 150 health workers that covered 61 questions and (ii) 12 in-depth interviews based on an internal interview guideline that has integrated relevant materials and opinions of experts in the field. The response rate for the survey was 100%.”
We have also discussed the limitations and implications of the sampling method toward the end of the paper (line 445-449).
“Moreover, with the convenience sampling method, the results presented in this paper should not be generalized to all healthcare facilities across Vietnam. As the study chose the hospitals/clinics in cosmopolitan areas in Vietnam and those who have worked with the MSM and transgenders before, one can reason that the misperceptions about these minorities might be much more common among the general population of healthcare workers.”
Final Thank-you Notes
In closing this letter, we would like to extend our greatest gratitude to the hard work and time that you have put into improving our manuscript. Thanks to your valuable feedback, we were able to revise our manuscript to be more coherent and relevant.
Thank you very much for your consideration. We hope that the findings and insights in this study will be shared and serving the academic community as a whole. Please accept our sincere thanks for your great contributions to the advancement of sciences in the world.
Shall you have further questions, please do not hesitate to contact us. We look forward to hearing from you.
Best regards,
The authors

Reviewer 2 Report
The authors evaluated the knowlegde about men who have sex with men (MSM) and transgenders for 150 health professionals in Vietnam using the questionnaire and in-depth interview for 12 participnats. The survey clearly demonstrated the high prevalence of misperceptions and lack of knowledge. The authors conclude the training courses should be prepared to facilitate understand the MSM and transgenders. The manuscript was well written with background og the study and current situation in Vietnam. It will be useful information for readers who are involved in not only gender clinic but also general health care through the world.
The reviewer would like to know the repsonse rate of this survey. Although the 150 persons answered, how many health professionals were initially asked to take part in this survey. If the response rate is very low (that may mean only health professionals who are familiar with MSM/transgenders and/or have the knowlegde about MSM/transgender were responded), is it possible that the real mispercepion is much more prevalent than the results in this survey?
Author Response
Dear Sir/Madam,
We would like to take this opportunity to express our deep appreciation for providing very helpful and detailed criticisms for our manuscript. Your meticulous input has helped us improve our manuscript significantly. In this revision, we have addressed your concerns in. Please notice that in the revised paper, the parts that are highlighted in yellow is for correction on the old text, the parts highlighted in green is written anew. Below are our answers to your comments (in italic). Also, the line numbers in the bold text refer to the revised paper.
The authors evaluated the knowledge about men who have sex with men (MSM) and transgenders for 150 health professionals in Vietnam using the questionnaire and in-depth interview for 12 participants. The survey clearly demonstrated the high prevalence of misperceptions and lack of knowledge. The authors conclude the training courses should be prepared to facilitate understand the MSM and transgenders. The manuscript was well written with background of the study and current situation in Vietnam. It will be useful information for readers who are involved in not only gender clinic but also general health care through the world.
The reviewer would like to know the response rate of this survey. Although the 150 persons answered, how many health professionals were initially asked to take part in this survey. If the response rate is very low (that may mean only health professionals who are familiar with MSM/transgenders and/or have the knowledge about MSM/transgender were responded), is it possible that the real misperception is much more prevalent than the results in this survey?
Thank you for your interest in our research. We sincerely apologize for our mistake in not including the response rate of the survey. We conducted the survey based on the convenience sampling method and the response rate was 100% (line 197-206).
“The survey was based on the convenience sampling method and these hospitals and clinics are chosen on the basis they already have some experienced working with MSM/transgenders. It is worth noting that the Vietnamese dermatology clinics/hospitals use the same system as the European system of combining training and treatment for dermato-venereology. This means that patient inquiries on venereology diagnosis and treatment can be answered by doctors at this type of hospital.
The respondents include physicians working in clinical settings and meeting patients on a daily basis as well as health professionals serving at leadership roles. In particular, the team carried out (i) a survey among 150 health workers that covered 61 questions and (ii) 12 in-depth interviews based on an internal interview guideline that has integrated relevant materials and opinions of experts in the field. The response rate for the survey was 100%.
We have also discussed the limitations as well as the implications of this method of sampling in the end of the paper.
“Moreover, with the convenience sampling method, the results presented in this paper should not be generalized to all healthcare facilities across Vietnam. As the study chose the hospitals/clinics in cosmopolitan areas in Vietnam and those who have worked with the MSM and transgenders before, one can reason that the misperceptions about these minorities might be much more common among the general population of healthcare workers.”
In closing this letter, we would like to extend our greatest gratitude to the hard work and time that you have put into improving our manuscript. Thanks to your valuable feedback, we were able to revise our manuscript to be more coherent and relevant. We hope that the findings and insights in this study will be shared and serving the academic community as a whole. Please accept our sincere thanks for your great contributions to the advancement of sciences in the world.
Shall you have further questions, please do not hesitate to contact us. We look forward to hearing from you.
Best regards,
